# Diagnostic Accuracy of Chest Digital Tomosynthesis in Patients Recovering after COVID-19 Pneumonia

Elisa Baratella [1,*,†] , Barbara Ruaro [2,†] , Cristina Marrocchio [1] , Gabriele Poillucci [1], Caterina Pigato [1], Alessandro Marco Bozzato [1] , Francesco Salton [2] , Paola Confalonieri [2] , Filippo Crimi [3] , Barbara Wade [4], Emilio Quaia [3,‡] and Maria Assunta Cova [1,‡]

1 Department of Radiology, Cattinara Hospital, University of Trieste, 34127 Trieste, Italy; cristinamarrocchio@gmail.com (C.M.); gabrielepoillucci@gmail.com (G.P.); pigatocaterina626@gmail.com (C.P.); alessandroj.bozzato@gmail.com (A.M.B.); m.cova@fmc.units.it (M.A.C.)
2 Department of Pulmonology, Cattinara Hospital, University of Trieste, 34127 Trieste, Italy; barbara.ruaro@yahoo.it (B.R.); francesco.salton@gmail.com (F.S.); paola.confalonieri.24@gmail.com (P.C.)
3 Department of Radiology, University of Padua, Via Nicolò Giustiniani, 2, 35128 Padua, Italy; filippo.crimi@unipd.it (F.C.); emilio.quaia@unipd.it (E.Q.)
4 AOU City of Health and Science of Turin, Department of Science of Public Health and Pediatrics, University of Torino, 10124 Torino, Italy; barbarawade@hotmail.com
* Correspondence: elisa.baratella@gmail.com; Tel.: +39-040399437
† These authors contributed equally to this work.
‡ These authors contributed equally to this work.

**Abstract:** Purpose: To assess the diagnostic accuracy of traditional chest X-ray (CXR) and digital tomosynthesis (DTS) compared to computed tomography (CT) in detecting pulmonary interstitial changes in patients having recovered from severe COVID-19. Materials and Methods: This was a retrospective observational study, and received local ethics committee approval. Patients suspected of having COVID-19 pneumonia upon emergency department admission between 1 March and 31 August 2020, and who underwent CXR followed by DTS and CT, were considered. Inclusion criteria were as follows: (1) patients with previous SARS-CoV-2 infection proven by a positive RT-PCR on nasopharyngeal swabs performed upon admission to the hospital, and with complete clinical recovery; (2) a diagnosis of SARS-CoV-2-related ARDS, according to the Berlin criteria, during hospitalization; (3) no recent history of other lung disease; and (4) complete imaging follow-up by CXR, DTS, and CT for at least 6 months and up to one year. Analysis of DTS images was carried out independently by two radiologists with 16 and 10 years of experience in chest imaging, respectively. The following findings were evaluated: (1) ground-glass opacities (GGOs); (2) air-space consolidations with or without air bronchogram; (3) reticulations; and (4) linear consolidation. Indicators of diagnostic performance of RX and digital tomosynthesis were calculated using CT as a reference. All data were analyzed using R statistical software (version 4.0.2, 2020). Results: Out of 44 patients initially included, 25 patients (17 M/8 F), with a mean age of 64 years (standard deviation (SD): 12), met the criteria and were included. The overall average numbers of findings confirmed by CT were GGOs in 11 patients, lung consolidations in 8 patients, 7 lung interstitial reticulations, and linear consolidation in 20 patients. DTS showed a significantly higher diagnostic accuracy compared to CXR in recognizing interstitial lung abnormalities—especially GGOs ($p = 0.0412$) and linear consolidations ($p = 0.0009$). The average dose for chest X-ray was 0.10 mSv (0.07–0.32), for DTS was 1.03 mSv (0.74–2.00), and for CT scan was 3 mSv. Conclusions: According to our results, DTS possesses a high diagnostic accuracy, compared with CXR, in revealing lung fibrotic changes in patients who have recovered from COVID-19 pneumonia.

**Keywords:** chest digital tomosynthesis; COVID-19; ARDS; follow-up

## 1. Introduction

Chest X-ray has been the mainstay of first-line imaging evaluation for suspected thoracic diseases for more than a century in most of the world, and remains so even today, despite advances in thoracic imaging technologies such as computed tomography (CT) and magnetic resonance imaging (MRI) [1–6].

Although chest X-ray (CXR) continues to provide vital medical information even today, there is an overlap of three-dimensional structures projected on a two-dimensional image, leading to a decreased contrast resolution, where abnormalities such as pulmonary nodules often go undetected even by the most skillful of radiologists [1–6].

Little is currently known as to the long-term pulmonary sequelae of patients who recover from SARS-CoV-2-related severe acute respiratory distress syndrome (ARDS) [2–11]. To date, no uniform criteria are available to guide the radiological evaluation of viral pneumonia in the context of a pandemic. Indeed, the choice of imaging techniques is based not only on the properties of the imaging techniques, but also on the resources of the individual hospital and, last but not least, ultimately depends on the judgement and experience of the team of professionals directly involved in the management of these patients [3–6].

Although chest X-ray (CXR) is still the mainstay for the assessment of patients with known or suspected COVID-19 pneumonia, its low sensitivity does not allow for the recognition of interstitial lung abnormalities [1,5]. It has also been reported that HRCT is excellent not only in the detection of acute pneumonia due to COVID-19, but also in the follow-up of patients who have long-term pulmonary impairments, including fibrotic-like changes [5–9].

Other studies have reported that digital tomosynthesis (DTS) provides more information as to small lung lesions and fine lung parenchyma reticulations than the conventional CXR, by removing overlapping clutter [12–16]. This implies that digital tomosynthesis (DTS) may well be an alternative imaging modality to CT for patient follow-up after COVID-19 pneumonia in the detection of lung parenchyma abnormalities.

This study aimed at assessing the diagnostic accuracy of DTS in detecting pulmonary interstitial changes in patients who had recovered from COVID-19 lung infection.

## 2. Materials and Methods

This retrospective study conformed to the principles of the Declaration of Helsinki for medical research, and the study protocol was approved by the institutional review board of our hospital. All patients admitted to the emergency department with suspected COVID-19 pneumonia between 1 March and 31 August 2020, who had CXR followed by DTS and high-resolution CT (HRCT), were initially identified from our hospital's electronic database. Patients were initially eligible for study inclusion if they met the following criteria: (1) patients with previous SARS-CoV-2 infection proven by a positive RT-PCR on nasopharyngeal swabs performed upon admission to the hospital, and with complete clinical recovery; (2) a diagnosis of SARS-CoV-2-related ARDS, according to the Berlin criteria, during hospitalization; (3) no recent history of other lung disease; and (4) 6 months to 1 year of imaging follow-up by CXR, DTS, and HRCT.

### 2.1. Chest X-ray

Firstly, CXR examinations were obtained using a digital radiography (Definium 8000; GE Healthcare, Chalfont St Giles, UK) system. The X-ray images were acquired with the patient in the upright position at the wall stand, with a focal spot size of 0.6 mm and a stationary anti-scatter grid (70 lines per cm; ratio 13:1).

### 2.2. DTS

The patients then underwent a DTS on the same day as the CXR, using a digital radiography system with tomosynthesis capability (Definium 8000; GE Healthcare, Chalfont St Giles, UK). This system has an X-ray tube (focal spot size 0.6 mm), a wall stand, a stationary anti-scatter grid (70 lines per cm; ratio 13:1), and a cesium iodide–amorphous silicon

(CsI/a-Si) indirect flat-panel detector (41 × 41 cm$^2$; 200 × 200 μm$^2$ pixel size), with CsI in a columnar structure. The X-rays were converted into light in this layer of thallium-doped CsI, and then the light was converted into electrical signals by Si photodiodes, and the signal was multiplexed to the readout electronics by thin-film transistors (TFTs) consisting of Si deposited on a glass substrate. The signal was digitized with 14-bit resolution (16,384 grey levels) by integrated electronics. Acquisition data were single linear sweeps of the X-ray tube over an angle of 36°, a voltage 120 kVp, a detector entrance dose of 0.5 μGy, a nominal focal spot 0.6 mm, an additional copper filtration of 0.1 mm, and a breath-hold acquisition time of 11 s. Fifty-four low-dose projections were acquired at regular angular intervals during the tube sweep, with a slice interval of 5 mm; then, images were reconstructed with a slice interval of 1 mm.

*2.3. CT Acquisition Protocol*

HRCT was performed from 13 to 17 days (average: 15 days) from DTS.

HRCT was performed with a 256-row multidetector CT system (Brilliance iCT 256, Philips, Best, the Netherlands). The patients were invited to hold their breath with tidal inspiration during scanning in the supine position. The technical parameters were as follows: a rotation time of 270 ms; a beam collimation of 128 × 2 × 0.625 mm; a normalized pitch of 0.975; a *z*-axis coverage of 160 mm; a reconstruction interval of 0.3 mm; a section reconstruction thickness of 1 mm; a tube voltage of 120 kV; a tube current (effective mA) of 280–400, depending on the patient's size; and a field of view of 40 cm. The CT images were analyzed at standard lung window settings (a window level of −600 HU and a window width of 1600 HU) and mediastinal window settings (window level 400–500 HU and a window width of 20–40 HU).

Image analyses of CXR and DTS examinations were carried out by two radiologists with 16 and 10 years of experience in chest imaging. During image analyses, both readers worked independently and were aware of the patients' identity and their clinical history, but were blinded to HRCT findings. The readers were allowed to use processing tools— such as windowing, image contrast, adjustment, or magnification—and to scroll the DTS images. All readings were performed on a picture archiving and communications system (PACS)-integrated workstation (19-inch TFT display, resolution 2560 × 1600 pixels) in a central location. The following imaging findings were evaluated: (1) ground-glass opacities (GGOs); (2) air-space consolidations with or without air bronchogram; (3) reticulations; and (4) linear consolidation.

*2.4. Statistical Analysis*

The sensitivity, specificity, positive and negative predictive value, and diagnostic accuracy of CXR and DTS were compared by both readers, keeping HRCT as the reference standard. Cohen's kappa statistics was calculated to assess inter-reader agreement. CXR and DTS results were compared using McNemar's test, with the level of significance set at $p < 0.05$. All data were analyzed using R statistical software (version 4.0.2, 2020).

## 3. Results

A total of 25/44 patients (17 males and 8 females) who had initially been identified (average age: 64 years—standard deviation (SD): 12) met the inclusion criteria, and were included in the study. HRCT identified ground-glass opacities in 11 patients, lung parenchyma consolidations in 8, lung interstitial reticulations in 7 (Figure 1), and linear consolidation in 20 (Figure 2). DTS had a higher diagnostic accuracy than CXR in recognizing lung abnormalities (Table 1). The inter-reader agreement was higher for DTS in the detection of GGO, consolidations, and reticulations (k = 0.71, 0.64, and 0.61 for DTS vs. k = 0.06, 0.35, and 0.36 for CXR, respectively), while it was lower for DTS compared to CXR in linear consolidation (k = 0.61 for DTS vs. k = 0.90 for CXR).

We compared the DTS and CXR results of the most experienced radiologist (Reader 1) for each radiological finding using McNemar's test. There was a statistically significant difference

between the two techniques in the identification of the linear consolidations ($p = 0.0009$) and GGO ($p = 0.0412$). For the other radiological signs, no statistically significant differences were found between DTS and CXR (consolidations $p = 1.000$; reticulations $p = 0.1336$).

The average effective dose was 0.10 mSv (0.07–0.32) for CXR, 1.03 mSv (0.74–2.00) for DTS, and 3 mSv (2–4) for CT.

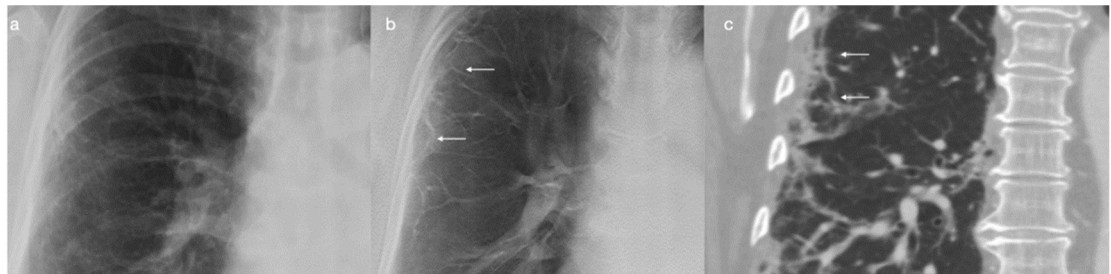

**Figure 1.** (**a**) A section of the follow-up CXR (posteroanterior view) of a 68-year-old female 3 months after COVID-19 pneumonia; (**b**) DTS clearly evidences subpleural reticulations, which were confirmed by CT ((**c**); coronal reconstruction, lung window).

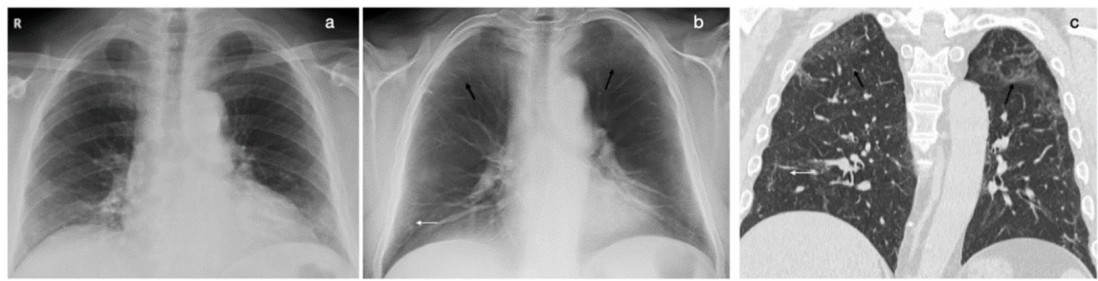

**Figure 2.** A 72-year-old male, 2 months after acute COVID-19 pneumonia: (**a**) The CXR, in the posteroanterior view, does not show any alterations. (**b**) The DTS evidences linear consolidations in the basal regions (white arrow) and subtle parenchymal opacities in the apical regions (black arrows); CT confirmed the DTS findings, evidencing ground-glass opacities in the apical regions ((**c**); coronal reconstruction, lung window).

**Table 1.** The diagnostic data provided by Readers 1 and 2 for CXR and DTS for each finding.

| CXR | GGO | | Consolidations | | Reticulations | | Linear Consolidations | |
|---|---|---|---|---|---|---|---|---|
| | Reader 1 | Reader 2 | Reader 1 | Reader 2 | Reader 1 | Reader 2 | Reader 1 | Reader 2 |
| Sensitivity (%) | 1/11 (9) | 2/11 (18) | 4/8 (50) | 3/8 (37) | 4/7 (57) | 4/7 (57) | 6/20 (30) | 7/20 (35) |
| Specificity (%) | 14/14(100) | 13/14 (93) | 16/17 (94) | 13/17(76) | 18/18(100) | 14/18 (78) | 5/5 (100) | 5/5 (100) |
| PPV (%) | 1/1 (100) | 2/3 (67) | 4/5 (80) | 3/7 (43) | 4/4 (100) | 4/8 (50) | 6/6 (100) | 7/7 (100) |
| NPV (%) | 14/24 (58) | 13/22 (59) | 16/20 (80) | 13/18(72) | 18/21 (86) | 14/17 (82) | 5/19 (26) | 5/18 (28) |
| Accuracy (%) | 15/25 (60) | 15/25 (60) | 20/25 (80) | 16/25 (64) | 22/25 (88) | 18/25 (72) | 11/25 (44) | 12/25 (48) |
| DTS | GGO | | Consolidations | | Reticulations | | Linear Consolidations | |
| | Reader 1 | Reader 2 | Reader 1 | Reader 2 | Reader 1 | Reader 2 | Reader 1 | Reader 2 |
| Sensitivity (%) | 5/11 (45) | 4/11 (36) | 6/8 (75) | 7/8 (87) | 7/7 (100) | 7/7 (100) | 19/20 (95) | 17/20 (85) |
| Specificity (%) | 12/14 (86) | 11/14 (79) | 17/17(100) | 16/17 (94) | 17/18 (94) | 16/18 (89) | 5/5 (100) | 5/5 (100) |
| PPV (%) | 5/7 (71) | 4/7 (57) | 6/6 (100) | 7/8 (87) | 7/8 (87) | 7/9 (78) | 19/19 (100) | 17/17 (100) |
| NPV (%) | 12/18 (67) | 7/18 (61) | 17/19 (89) | 16/17 (94) | 17/17 (100) | 16/16 (100) | 5/6 (96) | 5/8 (62) |
| Accuracy (%) | 17/25 (68) | 15/25 (60) | 23/25 (92) | 23/25 (92) | 24/25 (96) | 23/25 (92) | 24/25 (96) | 22/25 (88) |

Diagnostic performance of Reader 1 (R1) and Reader 2 (R2) in CXR and DTS for every finding category. Numbers between brackets are percentages. CXR = chest X-ray; DTS = digital tomosynthesis; GGO = ground-glass opacity; PPV = positive predictive value; NPV = negative predictive value.

## 4. Discussion

As patients recovering from severe COVID-19 pneumonia may require prolonged imaging follow-up of their persistent symptoms, it is essential to introduce diagnostic tools into clinical practice that meet the current needs dictated by this persisting pandemic [2,4–9].

We have demonstrated a superiority of DTS compared to CXR in the detection of "fibrotic-like" changes in the lung parenchyma after COVID-19 pneumonia—especially for linear consolidation ($p = 0.0009$) and GGO ($p = 0.0412$). Moreover, the average effective dose to which patients were exposed was lower for DTS compared to HRCT.

DTS does have some limitations, which include greater susceptibility to motion artifacts and limited depth resolution, making it more difficult to locate lesions near the diaphragm or chest wall and in subpleural locations, along with lack of portability [15,16]. Moreover, artifacts from respiration and medical devices such as ports and pacemakers have been reported [16]. Nevertheless, fiducial markers can correct patient motion and misregistration, and motion can be reduced by fast-scan DTS technology with a shorter breath hold, without losing diagnostic accuracy [17–19].

However, chest DTS, which has similar basic components to those of digital radiography, is also able to take advantage of some of the benefits that can be obtained by using computed tomography, although it is more time-consuming due to the large number of images produced (reported average = 200 s vs. 120 s) [15]. On the other hand, most CT coronal imaging of the chest requires image reconstruction; conversely, as a DTS system utilizes the full resolution of the detection panel, it does not require retrospective post-processing for coronal imaging [20,21].

Indeed, DTS has important advantages over conventional chest radiography, as it is able to provide better visibility of the pulmonary parenchyma, along with evidence of abnormalities such as pulmonary nodules, small lung lesions, and fine lung parenchyma reticulations, and allows for a better visualization of vessels, calcifications, airways, and chest wall abnormalities. Moreover, there are several other factors in favor of its adoption that should not be underestimated, in addition to its high diagnostic accuracy and inter-reader agreement. These include the fact that it is more "patient friendly"—as it uses a lower radiation dose than standard chest CT—and, last but not least, it is more economical [22,23].

Our study has some limitations that should be acknowledged. First of all, the small sample size could present an issue, but this should be considered as a "proof-of-concept" study for the application of DTS in the setting of the identification of chronic lung changes after severe COVID-19 pneumonia. Secondly, this study is retrospective and monocentric, although we are planning to develop a prospective multicentric study on the role of DTS in order to confirm our initial findings reported here. Moreover, a longer follow-up should probably have been indicated for these patients, so as to fully assess the real lung fibrotic residues of COVID-19 pneumonia.

According to the recent position paper from ESTI and ESR on imaging follow-up in patients recovering from COVID-19 pneumonia, patients who are not hospitalized and have no respiratory symptoms do not require any imaging follow-up. Meanwhile, patients with a severe primary infection complicated with acute respiratory distress syndrome (ARDS) who undergo mechanical ventilation show abnormalities on CT scan at 3-month follow-up, and these changes are more prominent in patients with a more severe primary infection. Thus, if radiological changes and respiratory symptoms persist in these patients, a radiological follow-up after a further 3 months is recommended; however, it is not reported whether these patients should undergo a chest X-ray or CT [24].

While short-term follow-up imaging findings are well described, there are only a few reports regarding the so-called "long-COVID" imaging findings, among which the incidence and the types of residual parenchymal alterations vary between studies, and there is no consensus concerning the exact timing of follow-up [25].

## 5. Conclusions

Briefly, although studies are ongoing to further validate its efficacy, we believe that DTS may well have a potential role to play as an alternative imaging tool to chest X-ray and CT in the monitoring of lung changes—particularly GGOs and "fibrotic-like" interstitial changes—and may even be able to replace them in many cases, aiding clinical workflow by increasing the availability of CT imaging, cutting costs, and respecting patient safety.

**Author Contributions:** Conceptualization, E.B. and G.P.; data curation, A.M.B.; writing—original draft preparation, E.B., B.R., C.M., G.P., and C.P.; writing—review and editing, B.R., F.S., P.C., F.C., and B.W.; supervision, E.Q. and M.A.C. All authors have read and agreed to the published version of the manuscript.

**Funding:** This research received no external funding.

**Institutional Review Board Statement:** This study was conducted according to the guidelines of the Declaration of Helsinki, and approved by the Local Ethical Committee—the institutional review board of Cattinara Hospital (protocol code (CEUR-2020-Os-148 and date of approval 30 September 2020).

**Informed Consent Statement:** Informed consent was obtained from all subjects involved in the study. The study was conducted according to the guidelines of the Declaration of Helsinki.

**Data Availability Statement:** All of the data are available upon reasonable request to the corresponding author.

**Conflicts of Interest:** The authors declare no conflict of interest.

## Abbreviations

HRCT: high-resolution computed tomography; DTS: digital tomosynthesis; CXR: chest X-ray; ARDS: acute respiratory distress syndrome.

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
