# Peer review of "Diagnostic Accuracy of Chest Digital Tomosynthesis in Patients Recovering after COVID-19 Pneumonia"

_tomography, doi:10.3390/tomography8030100_

Round 1

Reviewer 1 Report

Elisa et al. performed a very interesting clinical study, showing that Digital Tomosynthesis (DTS) could be a better diagnostic method in revealing lung fibrotic changes in patients who recovered from COVID-19 pneumonia compared with chest X-ray (CXR). The concern about the results is that DTS showed a significantly higher diagnostic accuracy than CXR in recognizing interstitial lung abnormalities. The p-value for GGOs is  p=0.0412, which is close to 0.05. The author probably needs to increase the number of penitents in the study to get a more solid statistical result. And here is another small issue: The authors need to explain the abbreviations: GG, GGO.

Author Response

We want to thank the reviewer for his/her helpful comment and for the possibility to resubmit the manuscript.

Elisa et al. performed a very interesting clinical study, showing that Digital Tomosynthesis (DTS) could be a better diagnostic method in revealing lung fibrotic changes in patients who recovered from COVID-19 pneumonia compared with chest X-ray (CXR).

The concern about the results is that DTS showed a significantly higher diagnostic accuracy than CXR in recognizing interstitial lung abnormalities. The p-value for GGOs is  p=0.0412, which is close to 0.05. The author probably needs to increase the number of penitents in the study to get a more solid statistical result.

R: We are totally agree with the reviewer comment.

Unfortunately, due to the retrospective nature of this study and the inclusion criteria (patients with a diagnosis of Sars-CoV-2 related ARDS during the first pandemic wave) we are not able to include other patients with similar characteristics.

And here is another small issue: The authors need to explain the abbreviations: GG, GGO.

R: We thank the reviewer for this comment. In agreement we have added the explanation in the text in the “Materials and Methods” section:

”(i) ground-glass opacities (GGOs)”

Reviewer 2 Report

Interesting manuscript but with some limitations, already reported by the authors.

It is not clear why the Chest Digital Tomosynthesis should be preferred over the classic CT exam.
For dose reduction?
But it must be demonstrated.
For a better patient workflow?
But this too must be specified.

Author Response

We want to thank the reviewer for his/her helpful comment and for the possibility to resubmit the manuscript.

Interesting manuscript but with some limitations, already reported by the authors.

It is not clear why the Chest Digital Tomosynthesis should be preferred over the classic CT exam.
For dose reduction? But it must be demonstrated. For a better patient workflow? But this too must be specified.

R: We thank the reviewer for these comments.

In accordance we have highlighted the DTS dose compare to CT in the “result” paragraph:

“The average effective dose was 0.10 mSv (0.07-0.32) for CXR, 1.03 mSv (0.74-2.00) for DTS and 3 mSv (2-4) for CT. “

Regarding dose reduction, It has been already demonstrated in numerous published papers that DTS provide a lower dose compared to CT [i.e. ref 22]

There another point that worth to be highlighted: the mean waiting time to perform a DTS is significantly lower compared to CT (i.e. in our Department the waiting time for a TDS in about 2-5 days while for a non-urgent CT scan can reach 20-30 days).

So DTS can be useful not only to reduce the dose but also for reducing the waiting time and the costs related to the overload of CT exams.

In accordance we have highlighted this point in the “discussion” paragraph and have added a related reference:

“These include the fact that it is more “patient friendly” as it uses a lower radiation dose than of standard chest CT and, last but not least, it is more economical [22, 23].”

Reviewer 3 Report

This is a study assessing the diagnostic accuracy of Digital Tomosynthesis (DTS) in detecting pulmonary interstitial changes in patients who had recovered from COVID-19 lung infection.

The manuscript is very well-written as a whole.  The authors admit that they have some limitations, but they are acceptable.

The English language is easy to understand except for the 3rd paragraph of Discussion section.  If “lack of portability” is one of the limitations of DTS, a comma should be added before “and lack of portability”.

It is understood that DTS is more accurate than CXR and safer than CT. Then, some questions can be raised as below:

  • Is DTS available in any countries?
  • Is DTS expensive to install?
  • Is special skill required to read DTS figure?

In my experience with COVID, patients with Delta suffered from interstitial pneumonia, but patients with the other variants did not progress into severe pneumonia.  I wonder what is the proportion of patients who require long-term follow-up after COVID pneumonia?  If they need follow-up for lingering aftereffects of COVID pneumonia, how long and how often do they need follow-up imaging?

This manuscript is acceptable for publication as it is, however it would be great if the authors can answer my questions.

Round 2

Reviewer 2 Report

The additions have enhanced the paper.

This manuscript is a resubmission of an earlier submission. The following is a list of the peer review reports and author responses from that submission.

Round 1

Reviewer 1 Report

I can understand your comments for my 2nd and 3rd pointed out parts. However, I can not accept the following points in the 1st part.

“Finally, in some cases the time between DTS and HRCT was longer than 1 month, thus, we cannot exclude that some radiological findings, especially GGOs, could have changed in that period.”

R: In cases 6 months to 1 year after diagnosis, does the one-month time difference between DTS and HRCT cause radiological changes in CT images? Why only GGO? If so, why can you say that the linear consolidation does not change? CT findings in HRCT are important points of your study in assessing accuracy. Your consideration of radiological findings also show that the assessment of accuracy in this study is incorrect.

Reviewer 2 Report

The suggested corrections were followed precisely. The paper appears clearer in intent and form as well as in the statistical part.

Author Response

We thank the Reviewer for this positive feedback.

Round 2

Reviewer 1 Report

The authors' comments are not an accurate response to the problems presented by the reviewer.

There was little justification for evaluating only the useful result of "Linear consolidation" and denying the negative result of "GGO".

If the one-month time difference between DTS and HRCT causes radiological changes in CT images, then a study using a comparison of imaging findings does not hold. I think that this research has a large bias even as a “proof-of-concept”.